# GREY-BOX EXTRACTION OF NATURAL LANGUAGE MODELS

## ABSTRACT

Model extraction attacks attempt to replicate a target machine learning model from predictions obtained by querying its inference API. An emerging class of attacks exploit algebraic properties of DNNs (Carlini et al., 2020; Rolnick & Körding, 2020; Jagielski et al., 2020) to obtain high-fidelity copies using orders of magnitude fewer queries than the prior state-of-the-art. So far, such powerful attacks have been limited to networks with few hidden layers and ReLU activations.

In this paper we present algebraic attacks on large-scale natural language models in a *grey-box* setting, targeting models with a pre-trained (public) encoder followed by a single (private) classification layer. Our key observation is that a small set of arbitrary embedding vectors is likely to form a basis of the classification layer's input space, which a grey-box adversary can compute. We show how to use this information to solve an equation system that determines the classification layer from the corresponding probability outputs.

We evaluate the effectiveness of our attacks on different sizes of transformer models and downstream tasks. Our key findings are that (i) with frozen base layers, high-fidelity extraction is possible with a number of queries that is as small as twice the input dimension of the last layer. This is true even for queries that are entirely in-distribution, making extraction attacks indistinguishable from legitimate use; (ii) with fine-tuned base layers, the effectiveness of algebraic attacks decreases with the learning rate, showing that fine-tuning is not only beneficial for accuracy but also indispensable for model confidentiality.

## 1 INTRODUCTION

Machine learning models are often deployed behind APIs that enable querying the model but that prevent direct access to the model parameters. This restriction aims to protect *intellectual property*, as models are expensive to train and hence valuable (Strubell et al., 2019); *security*, as access to model parameters facilitates the creation of adversarial examples (Laskov et al., 2014; Ebrahimi et al., 2018); and *privacy*, as model parameters carry potentially sensitive information about the training data (Leino & Fredrikson, 2020). Model extraction attacks (Tramèr et al., 2016) attempt to replicate machine learning models from sets of query-response pairs obtained via the model's inference API, thus effectively circumventing the protection offered by the API.

Several extraction attacks on deep neural networks (see Jagielski et al. (2020) for a recent overview) follow a *learning-based* approach (Tramèr et al., 2016; Orekondy et al., 2018; Pal et al., 2020; Krishna et al., 2020), where the target model is queried to label data used for training the replica. The replicas obtained in this way aim to achieve accuracy on the desired task, or agreement with the target model on predictions, but recovery of the model weights is out of scope of this approach.

More recently, a novel class of attacks has emerged that uses *algebraic* techniques to recover the weights of deep neural networks up to model-specific invariances. Examples are the attacks of Milli et al. (2018), which leverage observations of gradients to recover model parameters, and Rolnick & Körding (2020); Jagielski et al. (2020); Carlini et al. (2020), which estimate gradients from finite differences of logits, and then use this information to recover model parameters. Algebraic attacks improve on learning-based attacks in that they (i) achieve higher-fidelity replicas and (ii) are orders of magnitude more query-efficient. So far, however, algebraic attacks have only been applied to small, fully connected neural networks with ReLU activations. In particular, for modern large-scale

natural language models (LLMs) such as BERT or GPT-2, the state-of-the-art model extraction attack is still learning-based (Krishna et al., 2020).

In this paper, we propose the first algebraic attacks on LLMs. We focus on models consisting of a pre-trained encoder and a single task-specific classification layer. We assume a *grey-box* setting where the encoder is *public* (and hence known to the adversary), and the classification layer is *private* (and hence the main target of the attack).

There are two key observations that enable us to extract LLMs via algebraic attacks. The first is that it is sufficient for an adversary to *know* rather than to choose the embeddings that are fed into the last layer. Existing algebraic attacks can infer the inputs to hidden layers, but can only do so on piecewise linear networks and would not work on LLMs, which use non-linear activations. In the grey-box setting, an adversary can compute hidden embeddings of any input by querying the public encoder model, and can query the target LLM on the same input through the model's API. We show in theory, and confirm by experiments, that a random set of $n$ embeddings is likely to form a basis of the last layer's input space. The raw outputs (i.e., the *logits*) on this basis uniquely determine the parameters of the last linear layer, which can be recovered by a transformation to the standard basis.

Our second observation is that this approach extends to the case where the API returns *probabilities* rather than raw logits, after normalization by the *softmax* function. For this, we leverage the invariance under translation of *softmax* to establish an invariance result for linear functions. Using this result, we show that the parameters of the last layer can be recovered (up to invariance) from embedding vectors spanning its input space and their corresponding probability outputs.

We evaluate our attacks on LLMs of different sizes and fine-tuned to different downstream tasks. We study the effects of using different types and numbers of extraction queries and different learning rates for fine-tuning the encoder model. Our key findings are:

- When the target model's base layers are *frozen* during fine-tuning (i.e., the attacker can get the exact embedding of any input), the attack is extremely effective. With only twice as many queries as the dimension of the embedding space (e.g., 1536 for BERT-base), we extract models that achieve 100% fidelity with the target, for all model sizes and tasks.
- When the model's base layers are *fine-tuned* together with the task-specific layer, the embeddings of the base model only *approximate* those of the target model and, as expected, the fidelity of the extracted models decreases as the learning rate grows. Maybe surprisingly, for some models and downstream tasks, we are still able to extract replicas with up to 82% fidelity and up to 79% task accuracy, for orders of magnitude fewer queries than required by state-of-the-art learning-based attacks (Krishna et al., 2020).
- Extraction is possible using either random or in-distribution queries. Replicas extracted using in-distribution queries perform well on both in-distribution and random challenge inputs. This shows that replicas can be created from small numbers of in-distribution queries, making attempts to extract the model indistinguishable from legitimate use.

In summary, we propose a novel grey-box extraction attack on natural language models that is indistinguishable from legitimate use in terms of the content and number of queries required.

## 2 ATTACK

We consider classification models $h\colon X \to \mathbb{R}^n$, mapping elements from $X$ to label probabilities in $\mathbb{R}^n$. We assume that $h = \log \circ \operatorname{softmax} \circ f \circ g$ consists of three components

$$h\colon X \xrightarrow{g} \mathbb{R}^n \xrightarrow{f} \mathbb{R}^m \xrightarrow{\log \circ \operatorname{softmax}} \mathbb{R}^m$$

where

- $g\colon X \to \mathbb{R}^n$ is a contextualized embedding model, such as BERT or GPT-2;
- $f\colon \mathbb{R}^n \to \mathbb{R}^m$ is an affine function computing logits from embeddings, i.e., $f(x) = Ax + b$ with $A \in \mathbb{R}^{m \times n}$ and $b \in R^m$;
- $\operatorname{softmax}\colon \mathbb{R}^m \to \mathbb{R}^m$ normalizes logits to probability vectors:

$$\operatorname{softmax}(x) = \frac{\exp(x_i)}{\sum_{i=1}^{m} \exp(x_i)} . \tag{1}$$

We assume the adversary knows the embedding model $g$ and tries to infer $A$ and $b$ (resp. $f$). We call this adversary *grey-box* because it can compute the embeddings from the inputs to $h$.

## 2.1 BASIC IDEA

Milli et al. (2018) show how to reconstruct models from oracle access to gradients. Carlini et al. (2020) show how to replace gradients by finite differences of logits, enabling reconstructing models without such an oracle. To explain the basic idea, let $e_i = (0, \ldots 0, 1, 0, \ldots 0)^T$ be the $i$th vector of the standard basis in $\mathbb{R}^n$, and let $x \in \mathbb{R}^n$ be arbitrary. The $i$th column of $A$ can be obtained as the difference between $f(x + e_i)$ and $f(x)$. There are two obstacles that prevent us from directly applying this idea to classification layers from LLMs, namely (1) the attack requires the embeddings to be *chosen*, which would effectively amount to reversing the embedding model; and (2) the attack uses raw logits, while APIs provide only log probabilities normalized with *softmax*. We show next how to overcome these obstacles.

## 2.2 EXTRACTION FROM LOGITS

As a first step, we overcome the requirement that the adversary be able to *choose* the inputs to $f$. Specifically, we show that it is sufficient for the adversary to *know* the inputs, given that they are sufficiently random. We rely on the following standard result:

**Proposition 1.** *Let $x^{(1)}, \ldots, x^{(n)} \in \mathbb{R}^n$ be uniformly distributed in an $n$-cube. Then $x^{(1)}, \ldots, x^{(n)}$ form a basis of $\mathbb{R}^n$ with probability 1.*

*Proof.* Let $\{x^{(i)}\}_{i=1}^n$ be a basis of $\mathbb{R}^n$ and $V$ the subspace generated by $\{x^{(i)}\}_{i=1}^m$, for $m < n$. For $x$ chosen uniformly in an $n$-cube, the space $V \cup \{x\}$ has dimension $m+1$ with probability 1. This is because for $x$ to fall into $V$, it would need to have zero as coordinates wrt $x^{(m+1)}, \ldots, x^{(n)}$, which happens with probability zero. Applying this argument inductively proves the proposition. $\square$

Based on this result, we mount the following grey-box attack on $f \circ g$:

1. Choose distinct inputs $\{x^{(j)}\}_{j=1}^N$ in $X$ with $N > n$;
2. Compute their embeddings $\{y^{(j)} = g(x^{(j)})\}_{j=1}^N$ and logits $\{z^{(j)} = f(y^{(j)})\}_{j=1}^N$;
3. Construct a matrix $Y \in \mathbb{R}^{(n+1) \times N}$ where the first component of each column is 1 and the rest are from the embeddings, i.e., $Y_{i,j} = (1, y^{(j)})^T$, and a matrix $Z \in \mathbb{R}^{m \times N}$ where the columns are the logit vectors, i.e., $Z_{i,j} = z_i^{(j)}$;
4. Solve for $\tilde{A} \in \mathbb{R}^{m \times n}$ and $\tilde{b} \in R^m$ in $\left( \tilde{b}, \tilde{A} \right) Y = Z$, i.e.,

$$\begin{pmatrix} \tilde{b}_1 & \tilde{a}_{1,1} & \tilde{a}_{1,n} \\ \vdots & \ddots & \vdots \\ \tilde{b}_m & \tilde{a}_{m,1} & \tilde{a}_{m,n} \end{pmatrix} \begin{pmatrix} 1 & \cdots & 1 \\ y_1^{(1)} & \cdots & y_1^{(N)} \\ \vdots & \ddots & \vdots \\ y_n^{(1)} & \cdots & y_n^{(N)} \end{pmatrix} = \begin{pmatrix} z_1^{(1)} & \cdots & z_1^{(N)} \\ \vdots & \ddots & \vdots \\ z_m^{(1)} & \cdots & z_m^{(N)} \end{pmatrix} \quad (2)$$

**Proposition 2.** *Assuming $g$ maps inputs to uniformly distributed embeddings in an $n$-cube[1], the attack above uniquely determines the parameters of $f$. I.e., we have $\tilde{A} = A$ and $\tilde{b} = b$.*

*Proof.* By construction of (2), we have $\tilde{A}y^{(j)} + \tilde{b} = f(y^{(j)})$ for $j = 1, \ldots, N$. The unique solution can be obtained multiplying $Z$ by the right-inverse of $Y$. For uniformly random embeddings, this right-inverse exists because $Y$ has full rank with probability 1 by Proposition 1. $\square$

While in theory $N = n+1$ distinct queries are sufficient to mount the attack, computing the inverse of $Y$ based with finite-precision arithmetic can be numerically unstable. In practice, we gather a larger set of inputs and construct an over-determined system of equations. We then numerically compute a least squares solution to Equation (2).

---

[1] A language model with vocabulary $V$ and maximum sequence length $L$ can only produce $|V|^L$ different embeddings. This is many more points than representable in the precision used, so not an issue in practice.

## 2.3 EXTRACTION FROM LOG PROBABILITIES

We have so far assumed that the adversary has access to the raw unnormalized values, or logits. We next extend the attack to the case where the target model's inference API exposes only the log probability of each class, obtained by normalizing logits using *softmax* and returning the component-wise log of the result.

*Softmax* is invariant under translation, i.e., $\mathrm{softmax}(x) = \mathrm{softmax}(x + (c, \ldots, c))$. We lift this property to linear functions:

**Proposition 3.** *Let $C \in \mathbb{R}^{m \times n}$ be a matrix with identical rows, i.e., $c_{i,j} = c_{i',j}$ for all $i, i', j$ within range. Then for all $A \in \mathbb{R}^{m \times n}$ and $y \in \mathbb{R}^n$:*

$$\mathrm{softmax}((A + C)y) = \mathrm{softmax}(Ay)$$

*Proof.* Observe that $\mathrm{softmax}((A + C)y) = \mathrm{softmax}(Ay + (k, \ldots, k)^T) = \mathrm{softmax}(Ay)$, where $k = \sum_{j=1}^{n} c_{1,j} y_j$ since all rows of $C$ are identical. The last equality follows from the translation invariance of $\mathrm{softmax}$. □

Due to Proposition 3 we cannot hope to recover $A$ and $b$ (as in Proposition 2). However, we can still obtain weights that yield functionally equivalent results when post-processed with *softmax*. Based on the construction of $C$ above, we propose the following grey-box attack on $h$:

1. Gather inputs and construct a matrix $Y \in \mathbb{R}^{(n+1) \times N}$ as described in Steps 2 and 3 in Section 2.2.
2. Define $D \in \mathbb{R}^{(m-1) \times N}$ such that $D_{i,j} = p_i^{(j)} - p_1^{(j)}$, where $p^{(j)} = \log(\mathrm{softmax}(f(y^{(j)})))$ are the log probabilities of the inputs. That is, we collect differences of log probability vectors as columns in $D$, and then subtract the first row from all rows.
3. Define $\tilde{b}_1 = 0$ and $\tilde{a}_{1,j} = 0$ for $j = 1, \ldots, n$. Solve for the remaining components of $\tilde{b}$ and $\tilde{A}$ from

$$\begin{pmatrix} \tilde{b}_2 & \tilde{a}_{2,1} & \tilde{a}_{2,n} \\ \vdots & \ddots & \vdots \\ \tilde{b}_m & \tilde{a}_{m,1} & \tilde{a}_{m,n} \end{pmatrix} \begin{pmatrix} 1 & \cdots & 1 \\ y_1^{(1)} & \cdots & y_1^{(N)} \\ \vdots & \ddots & \vdots \\ y_n^{(1)} & \cdots & y_n^{(N)} \end{pmatrix} = \begin{pmatrix} p_2^{(1)} - p_1^{(1)} & \cdots & p_2^{(N)} - p_1^{(N)} \\ \vdots & \ddots & \vdots \\ p_m^{(1)} - p_1^{(1)} & \cdots & p_m^{(N)} - p_1^{(N)} \end{pmatrix} \quad (3)$$

**Proposition 4.** *The attack determines the parameters of $f(x) = Ax + b$ up to translation, that is:*

$$\mathrm{softmax}(\tilde{A}y + \tilde{b}) = \mathrm{softmax}(Ay + b)$$

For a proof, observe that $p_i^{(k)} = \sum_{j=1}^{n} a_{i,j} y_j^{(k)} + b_i - \log\left(\sum_{i=1}^{m} \exp\left(z_i^{(k)}\right)\right)$ where $z^{(k)} = Ay^{(k)} + b$. Hence we have, for $i = 2, \ldots, m$:

$$Y_{i,k} = p_i^{(k)} - p_1^{(k)} = \sum_{j=1}^{n} (a_{i,j} - a_{1,j}) y_j^{(k)} + (b_i - b_1) \quad (4)$$

By construction we have $\tilde{a}_{i,j} = a_{i,j} - a_{1,j}$ and $\tilde{b}_i = b_i - b_1$, which implies that the columns of $A - \tilde{A}$ and $b - \tilde{b}$ are constant vectors (resp. the rows of $A - \tilde{A}$ are all identical). From Proposition 3, we finally obtain

$$\mathrm{softmax}(\tilde{A}y + \tilde{b}) = \mathrm{softmax}((A - (A - \tilde{A}))y + (b - (b - \tilde{b}))) = \mathrm{softmax}(Ay + b)$$

As before we can gather more than $n + 1$ inputs and build an solve the over-determined system (3) to leverage more queries and improve numerical robustness.

## 2.4 EXTRACTION USING APPROXIMATIONS OF EMBEDDINGS

The attacks presented in Sections 2.2 and 2.3 rely on the assumption that the adversary has full access to the public embedding model $g$. In many settings, the embedding model is fine-tuned together with the task-specific layer $f$, and so the target model of the attack is $h' = \log \circ \mathrm{softmax} \circ$

$f \circ g'$, rather than $h = \log \circ \operatorname{softmax} \circ f \circ g$. We extend our attack to such settings by using the public embedding model $g$ as an approximation of the embedding model $g'$. That is, we do *not* assume that the adversary knows $g'$; instead we attack $h'$ based on embeddings from $g$, which is why the function $f^*$ we extract is only an approximation of $f$. We then construct the replica $h^* = \log \circ \operatorname{softmax} \circ f^* \circ g$ by wiring the extracted classification layer on top of the public encoder.

Clearly, the performance of this approach depends on the quality of the approximation $g \approx g'$, which is likely to decrease with the learning rate $\eta$ with which $g$ is transformed into $g'$. In the next section we evaluate the feasibility of this approach for different learning rates $\eta$.

Even if the attacker does not know *which* embedding model was used (i.e., a fully black-box attack), it is still possible to perform a variant of our attack. The attacker can collect query-response pairs from the target model (since these do not depend on the choice of embedding model) and run the attack (offline) using the same pairs but different embedding models. In practice, there are relatively few widely-used embedding models, so the attacker can perform a brute-force exploration.

## 3 EXPERIMENTS

We evaluate our attack on models for text classification from the GLUE benchmark: SST-2 (Socher et al., 2013) and MNLI (Williams et al., 2017). SST-2 is a binary sentiment analysis task where the goal is to predict positive or negative sentiment of an input sentence. MNLI is a task with 3 output classes to predict a relation between a premise and a hypothesis.

We train different target models using two base models: 1) BERT-Base with 12 layers and 768-dimensional embeddings and 2) BERT-Small with 4 layers and 512-dimensional embeddings. We vary the learning rate of the base model ($\eta$ ranges from 0 to $2 \times 10^{-5}$) while the classifier layer is always trained with a fixed learning rate $\eta = 2 \times 10^{-5}$. In this section, all references to *learning rate* refer to the learning rate of the base layers. All our models are trained for 3 epochs using Hugging Face Transformers v.3.2.0 [2]. Our core attack logic is simple and is implemented in only 15 lines of Python code with around 500 lines of boilerplate.

To perform our attack, we vary the type (in-distribution vs. random) and number of queries made to the target model for extraction of the classification layer. For the real queries, we use the respective SST-2 and MNLI *test* sets, since these are in-distribution but unseen during training, and for the random queries we generate random strings (or pairs of strings for MNLI) of varying length up to the maximum length of the model (128). For attack evaluation, we use the public BERT model (small or base depending on the target model) and combine it with the extracted classification layer to form a complete extracted model (as mentioned in Section 2.4). For each extracted model, we measure the accuracy and agreement with the target model on the validation set of the task or different random *challenge* inputs, respectively.

**Research Questions.** We experimentally evaluate our attack to answer the following questions:

- **Type of extraction queries:** How does the type of query submitted by the attacker (e.g., in-distribution vs. random) affect the utility of the extracted model.
- **Number of extraction queries:** How does the number of queries used by the attacker affect the accuracy and agreement of the extracted model as compared to the target model?
- **Effect of base learning rate:** How does fine-tuning of the base model with different learning rates impact the success of our attack?

**Main Results.** Table 1 summarizes the key results across our evaluation for two extreme cases where (i) the base layers of the target model are frozen ($\eta = 0$) or (ii) fine-tuned with the same learning rate as the classifier layer ($\eta = 2 \times 10^{-5}$). Our findings are:

- For frozen base models, the attack produces models with 100% agreement, with numbers of queries equal (for SST-2) or twice as large (for MNLI) as the dimension of the embedding vector.
- For the fine-tuned models, agreement drops to 10% below the learning-based state-of-the-art (Krishna et al., 2020) for SST-2, but is achieved with an order of magnitude less queries: 1821 versus 67 349.

---

[2]https://github.com/huggingface/transformers

Table 1: Key results for different model sizes and downstream tasks. Agreement shows on how many inputs the extracted model exactly matches with the target model. #queries denote the number of queries required to extract the model. For SST-2, we are limited by the test inputs available in the dataset i.e., 1821. $H$ is the output dimension of the base model.

| Dataset | Base Model | Frozen ($\eta = 0$) | | | Fine-tuned ($\eta = 2e-5$) | | |
|---|---|---|---|---|---|---|---|
| | | Target Acc. | Agree-ment | #queries | Target Acc. | Agree-ment | #queries |
| SST-2 | BERT-base | .75 | 1.0 | $769\ (H+1)$ | .91 | .83 | $1821\ (2.3*H)$ |
| | BERT-small | .70 | 1.0 | $513\ (H+1)$ | .87 | .79 | $1821\ (3.5*H)$ |
| MNLI-3 | BERT-base | .43 | 1.0 | $1024\ (2*H)$ | .83 | .44 | $3456\ (4.5*H)$ |
| | BERT-small | .45 | 1.0 | $1536\ (2*H)$ | .77 | .44 | $3584\ (7*H)$ |

Table 2: Effect of using real vs. random queries for extraction ($\eta = 2 \times 10^{-5}$, #queries = 2H). $L^{\infty}A$ and $L^{\infty}b$ denote the difference between the target and extracted matrices. Agreement is computed for both in-distribution real inputs and randomly generated inputs.

| Dataset | Base Model | Extraction with Real queries | | | | Extraction with Random queries | | | |
|---|---|---|---|---|---|---|---|---|---|
| | | $L^{\infty}A$ | $L^{\infty}b$ | Agreement | | $L^{\infty}A$ | $L^{\infty}b$ | Agreement | |
| | | | | Real | Rand. | | | Real | Rand. |
| SST-2 | BERT-base | 85 | 155 | 0.81 | 0.89 | 21 | 3 | 0.63 | 0.90 |
| | BERT-small | 881 | 601 | 0.74 | 0.76 | 12 | 40 | 0.67 | 0.86 |
| MNLI-3 | BERT-base | 494.05 | 56.11 | .41 | .21 | 30.37 | 14.01 | 0.36 | 0.74 |
| | BERT-small | 1704.2 | 1032.98 | 0.48 | 0.39 | 80.84 | 99.97 | 0.33 | 0.88 |

- The attack works irrespective of the size of the base model, and performs better on down-stream tasks with fewer output classes.

**Effect of type of queries: Real vs. Random.** The type of queries used to extract the model could impact both the agreement of the extracted model, as well as the defender's ability to detect the attack. To explore this effect, we perform extraction attacks using both real (i.e., in-distribution) queries, or randomly-generated queries. In all cases, we use only $2H$ extraction queries (i.e., 1536 for BERT-Base and 1024 for BERT-Small models). We evaluate the worst-case scenario (from an attacker's perspective) where the base layers have been fine-tuned with the same learning rate as the classification layers ($\eta = 2 \times 10^{-5}$). The results are shown in Table 2 and the key observations are:

- A model extracted using real queries provides a similar level of agreement with the target model on both real and random inputs. Thus, extraction using real queries works better in general and is harder to distinguish from genuine benign queries.
- A model extracted using random queries provides better agreement with the target model on random inputs than on real inputs. The gap between agreement on real and random inputs is more pronounced for this type of model.
- $L^{\infty}$ distances between the target and extracted weight matrices and bias vectors are large, but do not necessarily affect agreement due to the invariance of the softmax operation.

**Effect of number of queries.** Using real queries, we vary the number of queries used for the extraction and report both the task accuracy of the extracted model and the agreement between the target and extracted models in Figure 1. The size of the respective test sets limits the number of queries we could use for this experiment. Again we use a learning rate of $\eta = 2 \times 10^{-5}$ to evaluate the worst-case scenario from the attacker's perspective. Note that the baseline task accuracy for a model performing random guessing on a balanced dataset for SST-2 and MNLI is 50% and 33% respectively. The key observations are:

- For both tasks and models, our extracted model performs better than a random guess and we observe clear increase in extracted model task accuracy and agreement as queries increase.
- After a sharp initial increase, there appear to be diminishing returns beyond $2H$ queries (i.e., 1536 for BERT-Base and 1024 for BERT-Small models).

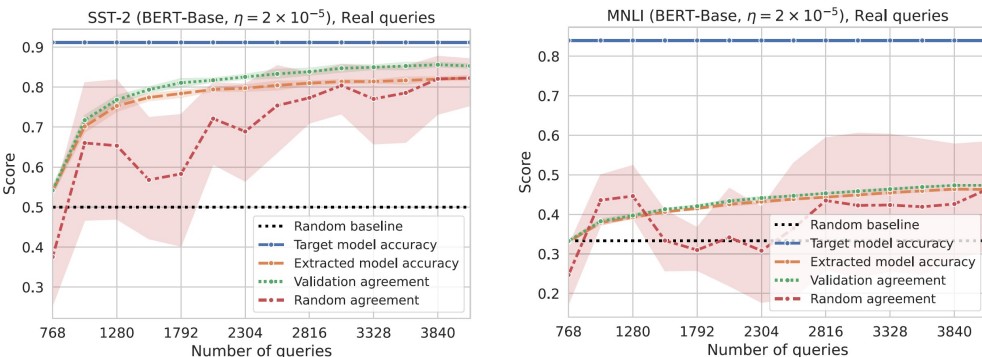

Figure 1: Effect of number of in-distribution queries on extracted model accuracy and agreement with the target model. Full results (including BERT-Small and random queries) in Appendix C.

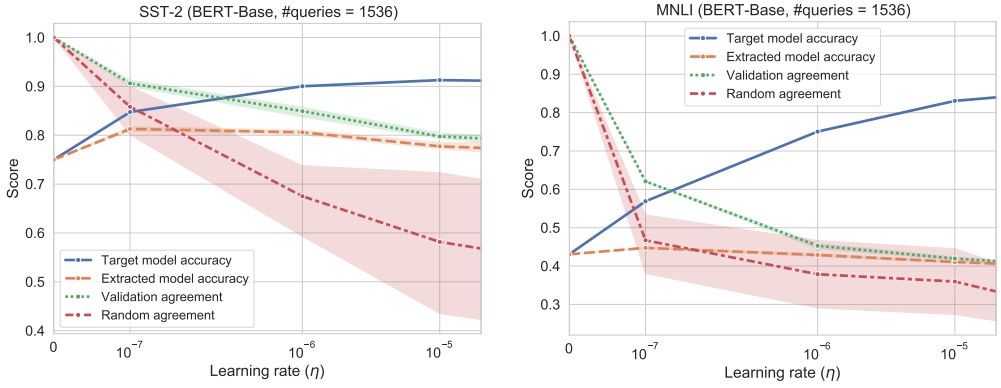

Figure 2: Effect of learning rate on target and extracted model accuracy, and agreement with the target model. Full results (including BERT-Small and random queries) in Appendix B.

- As above, we achieve lower absolute task accuracy and agreement for MNLI than for SST-2, since the number of output classes increases.

Overall, the number of queries we require to achieve reasonable agreement is still orders of magnitude lower than prior work.

**Effect of learning rate.** Finally, we quantify the effect of the learning rate used to fine-tune the base layers. We attack target models trained with learning rates ranging from $0$ (frozen) to $2 \times 10^{-5}$ (note that the classification layers all use $2 \times 10^{-5}$) using real queries. The accuracies and agreement of target and extracted models are depicted in Figure 2. We highlight the following:

- Agreement between the target and extracted models always starts at 100% for frozen base layers but decreases with increase in learning rate.
- As expected, original accuracy increases with learning rate and, interestingly, extracted model accuracy also increases slightly initially before decreasing for higher learning rates.

## 4 DISCUSSION

**Defenses against model extraction attacks.** Several defences against model stealing attacks focus on identifying malicious query patterns. Juuti et al. (2019); Atli et al. (2020) collect stateful information about queries at the API-level and flag potential attacks as deviations from a benign distribution. Kariyappa & Qureshi (2019) propose a defense that selectively returns incorrect predictions for out-of-distribution queries. While such approaches are shown to be effective against learning-based attacks, our attack can leverage random and in-distribution queries alike, and will hence evade such defenses. Other defenses rely on limiting the information available to the adversary, for example by

quantizing prediction probabilities (Tramèr et al., 2016), or adding perturbations (Lee et al., 2018) to poison the attacker's training objective (Orekondy et al., 2020), or watermarking the model so that extraction becomes detectable (Uchida et al., 2017). We expect these kinds of defenses may be effective against algebraic attacks, but leave an in-depth investigation as future work.

**Further improving the extracted model.** An attacker could combine our attack with techniques from existing learning-based extraction attacks. For example, after extracting the classification layer and adding this to a public embedding embedding model (as described in Section 2.4), the attacker could fine-tune this new model using the set of query-response pairs originally collected for our extraction attack, as well as any further query-response pairs from the target model. We investigate this hybrid attack strategy and its converse in Appendix A but leave a more in depth evaluation to a later revision.

## 5 RELATED WORK

There is a growing body of work studying the extraction of machine learning models, see e.g.,Lowd & Meek (2005); Tramèr et al. (2016); Orekondy et al. (2018); Rolnick & Körding (2020); Pal et al. (2020); Krishna et al. (2020); Carlini et al. (2020). These approaches differ in terms of the adversary's objectives, the model architecture, and the techniques they employ for extraction, see Jagielski et al. (2020) for a recent taxonomy and survey. For conciseness we focus this discussion on work that targets natural language model or uses techniques related to ours, as well as on defenses.

**Extraction of Natural Language Models.** Krishna et al. (2020) are the first to report on model extraction of large natural language models. They rely on a learning-based approach where they use the target model to label task-specific queries, which they craft based on random words. They also observe that transfer learning facilitates model extraction in that the adversary can start with a known base model for training the replica. Our attack goes one step further in that we leverage public knowledge about the embeddings for mounting an algebraic attack on the last layer. Krishna et al. (2020) report on agreement (accuracy) of $0.92$ $(0.90)$ for SST for around 60K random queries on SST2, and of $0.80$ $(0.76)$ for $392\,702$ random queries for MNLI. In contrast, our attack requires only $2 * H$ (1024 and 1536 for BERT-Small and BERT-Base) number of queries with $H$ being the dimension of the base-embedding model.

**Algebraic model extraction attacks.** Most model extraction attacks rely on training the replica on labels generated by the target model. Recently, a class of attacks has emerged that uses algebraic techniques to recover deep neural networks, achieving copies with higher fidelity using smaller numbers of queries as compared to learning-based approaches. The core idea goes back to Milli et al. (2018), which leverage observations of gradients to recover model parameters. This is leveraged by Rolnick & Körding (2020); Jagielski et al. (2020); Carlini et al. (2020) which estimate gradients from finite differences of logits, and then use this information to recover model parameters.

Our attack differs from these approaches in different aspects. First, we only extract a single layer, whereas the other attacks have been demonstrated for up to 2 hidden layers. Second, as our attack is grey-box, we only assume that the attacker *knows* the inputs, whereas the other approaches require that the adversary be able to *choose*. Third, we show how to extract the model despite a *softmax* activation layer, which is out of scope of the other approaches.

**Relationship to softmax regression.** The problem of extracting models where only task-specific layers are fine-tuned is closely related, but not equivalent, to parameter estimation for softmax regression, see, e.g., Van der Vaart (2000); Yao & Wang (2019). The key difference is that for extraction the goal is to recover a fixed but unknown set of parameters (i.e. a ground truth) with a minimal amount of data, whereas for regression the goal is to find the best parameters to fit the data.

## 6 CONCLUSION

In conclusion, we propose a novel grey-box extraction attack on natural language models that is indistinguishable from legitimate use in terms of the content and number of queries required. Existing detections and defenses based on the number or type of queries are unlikely to be effective, and thus other approaches are needed to detect or mitigate grey-box extraction attacks.

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

Table 3: Results for learning-based (Distill), algebraic attacks (Extract), and hybrid attacks using an extraction dataset 10% the size of the training dataset of the target model (6734 queries for SST-2 and 39 270 for MNLI). All layers of the target model were fine-tuned with learning-rate $2 \times 10^{-5}$. We use the same hyperparameters as Krishna et al. (2020) for distillation.

| Task | Model (Acc.) | Extract | | Distill | | Extract-then-Distill | | Distill-then-Extract | |
|------|--------------|---------|------|---------|------|---------------------|------|---------------------|------|
| | | Acc. | Agr. | Acc. | Agr. | Acc. | Agr. | Acc. | Agr. |
| SST-2 | BERT-Small (0.875) | 0.751 | 0.803 | **0.856** | 0.928 | 0.806 | 0.849 | 0.846 | **0.930** |
| | BERT-Base (0.912) | 0.827 | 0.858 | 0.885 | 0.919 | 0.907 | **0.945** | **0.909** | 0.936 |
| MNLI | BERT-Small (0.777) | 0.479 | 0.504 | 0.724 | 0.854 | 0.660 | 0.741 | **0.731** | **0.862** |
| | BERT-Base (0.840) | 0.496 | 0.515 | 0.803 | 0.888 | 0.763 | 0.825 | **0.806** | **0.894** |

## A  COMBINING LEARNING-BASED AND ALGEBRAIC ATTACKS

We draw a side-by-side comparison between learning-based and algebraic extraction attacks and explore whether combining both types of attacks can give better overall results. We follow the learning-based approach from Krishna et al. (2020), which uses (black-box) distillation to extract an approximation of a target model. We evaluate two different hybrid attacks:

1. Distill-then-Extract. First distill a model using the public pre-trained encoder as a base. Then use algebraic extraction reusing the same set of queries to extract and replace the last layer.

2. Extract-then-Distill. First extract the last layer using the algebraic attack with the public pre-trained encoder. Then reuse the same set of queries to distill the resulting model.

The results are shown in Table 3. They show that although learning-based attacks on their own do generally better than algebraic attacks for large number of queries, their combination using either strategy achieves consistently better agreement. The Distill-then-Extract strategy appears superior to the Extract-then-Distill strategy. Given that an algebraic attack reusing queries from distillation is inexpensive and that it starts from a good baseline, the gains although modest show that combining both attacks is cheap and effective.

## B  FULL EXPERIMENTAL RESULTS WITH VARYING LEARNING RATE

Effect of learning rate on task accuracy of extracted models, and agreement with target model on in-distribution (Fig. 3) and random (Fig. 4) queries.

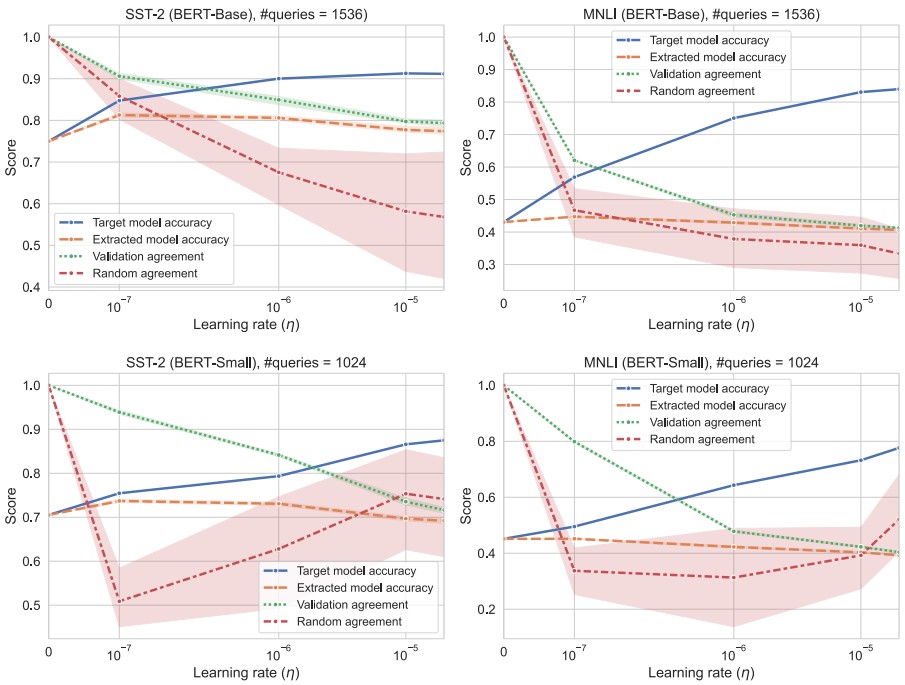

Figure 3: Extraction with in-distribution queries

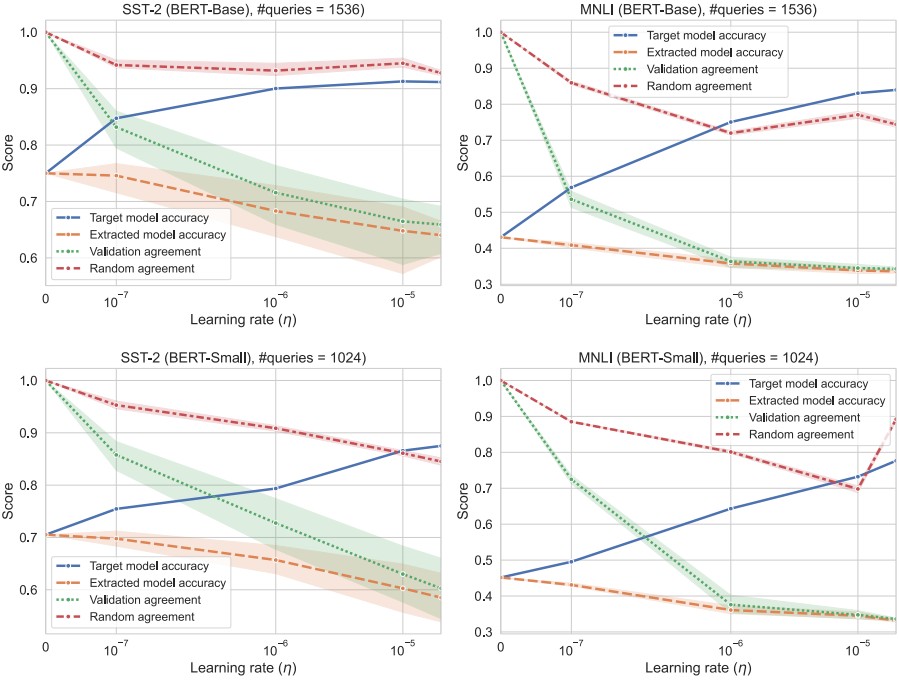

Figure 4: Extraction with random queries

# C    FULL EXPERIMENTAL RESULTS WITH VARYING NUMBER OF QUERIES

## C.1    BERT-BASE AND SST-2

Effect of number of queries on task accuracy of extracted model and agreement with target model, for in-distribution (Fig. 5) and random (Fig. 6) queries. Baseline accuracy of random guess: $50\%$.

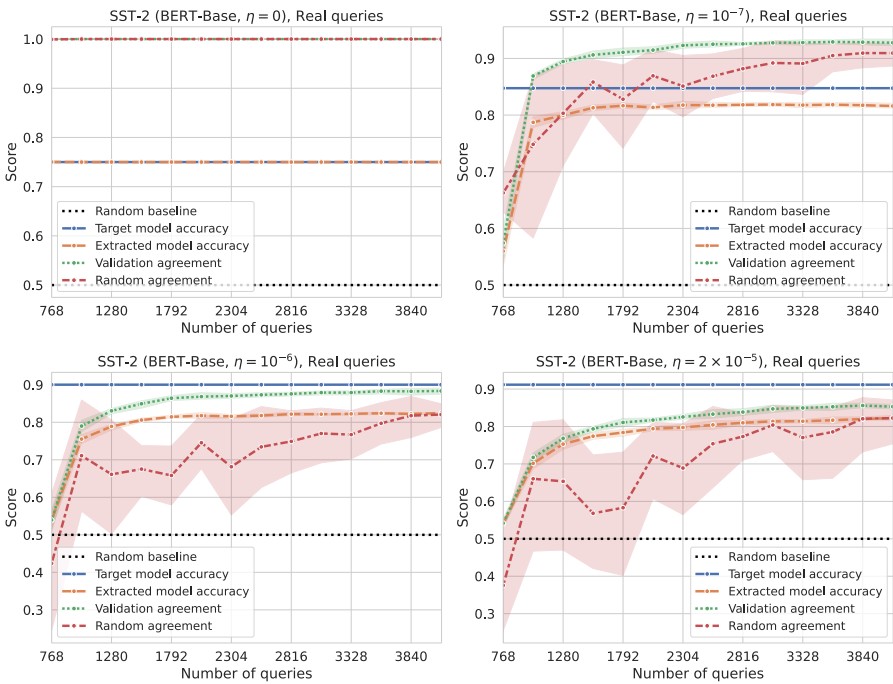

Figure 5: Extraction with in-distribution queries

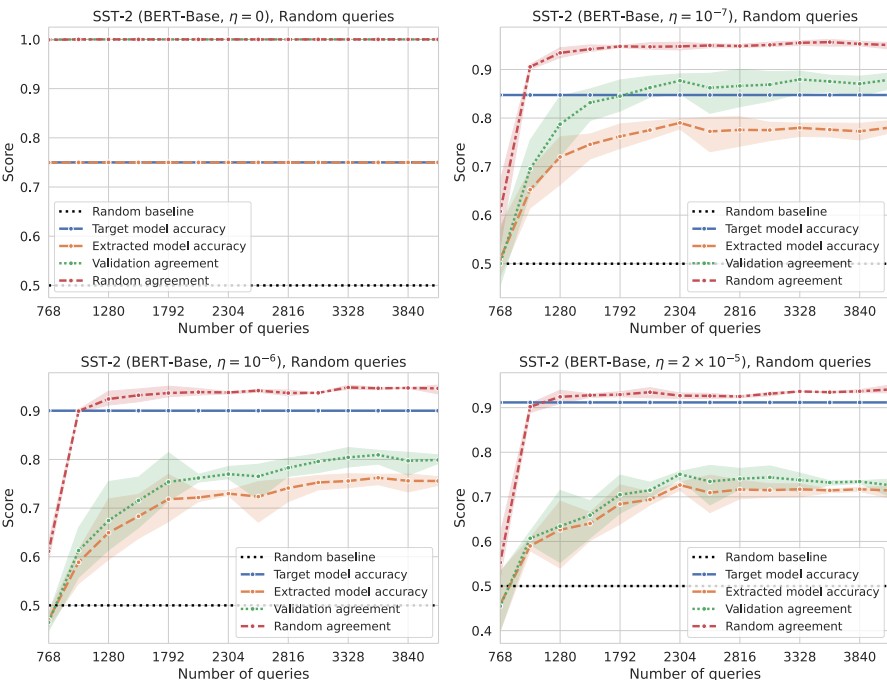

Figure 6: Extraction with random queries

## C.2 BERT-BASE AND MNLI

Effect of number of queries on task accuracy of extracted model and agreement with target model, for in-distribution (Fig. 7) and random (Fig. 8) queries. Baseline accuracy of random guess: 33%.

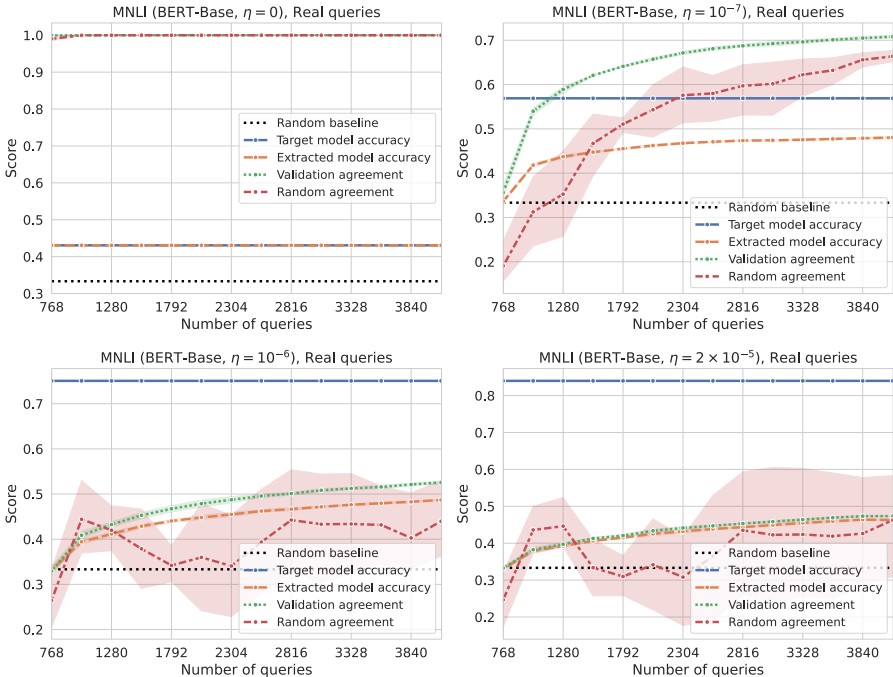

Figure 7: Extraction with in-distribution queries

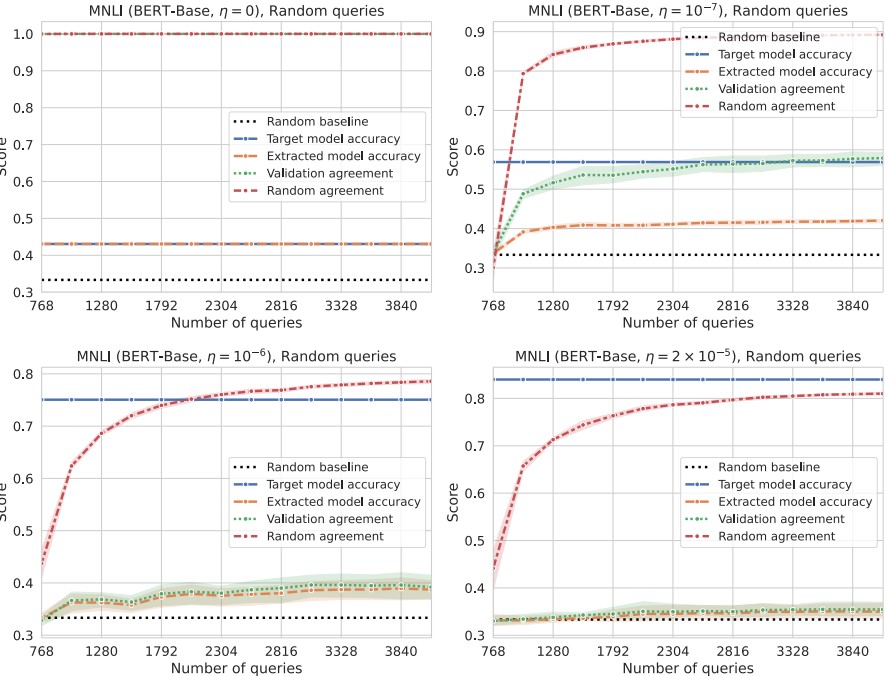

Figure 8: Extraction with random queries

## C.3 BERT-SMALL AND SST-2

Effect of number of queries on task accuracy of extracted model and agreement with target model, for in-distribution (Fig. 9) and random (Fig. 10) queries. Baseline accuracy of random guess: $50\%$.

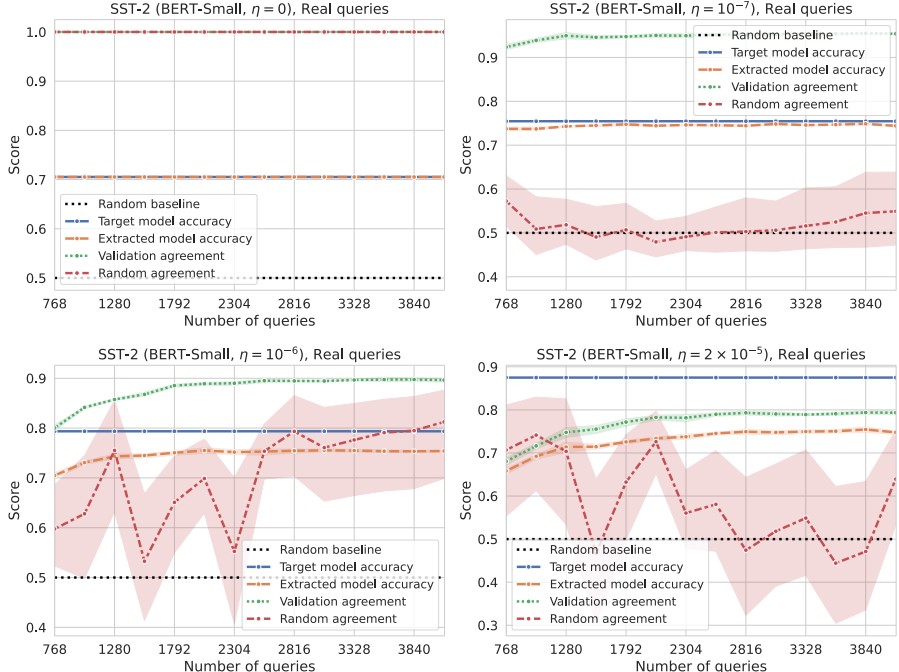

Figure 9: Extraction with in-distribution queries

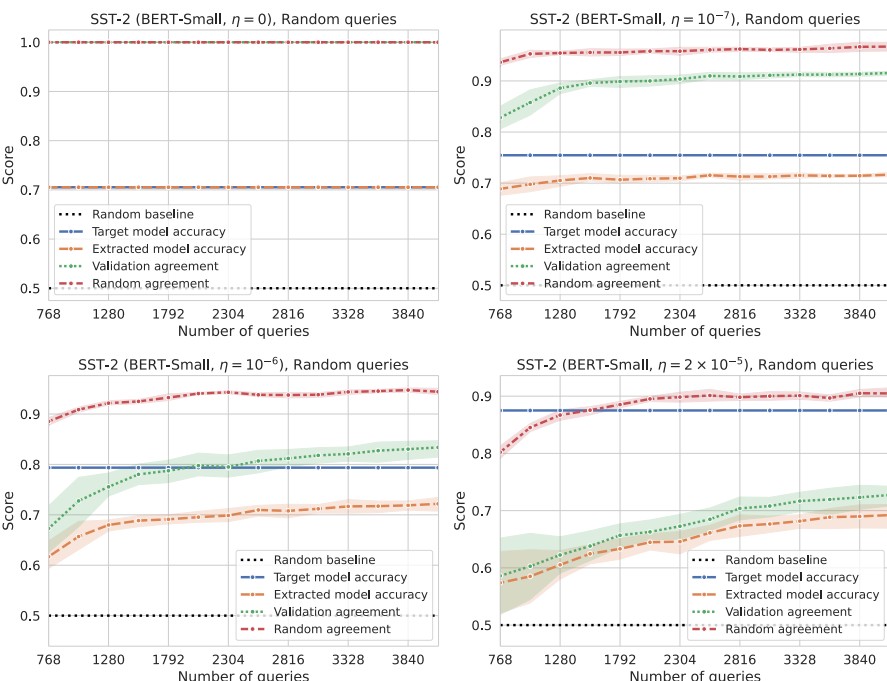

Figure 10: Extraction with random queries

## C.4 BERT-SMALL AND MNLI

Effect of number of queries on task accuracy of extracted model and agreement with target model, for in-distribution (Fig. 11) and random (Fig. 12) queries. Baseline accuracy of random guess: $33\%$.

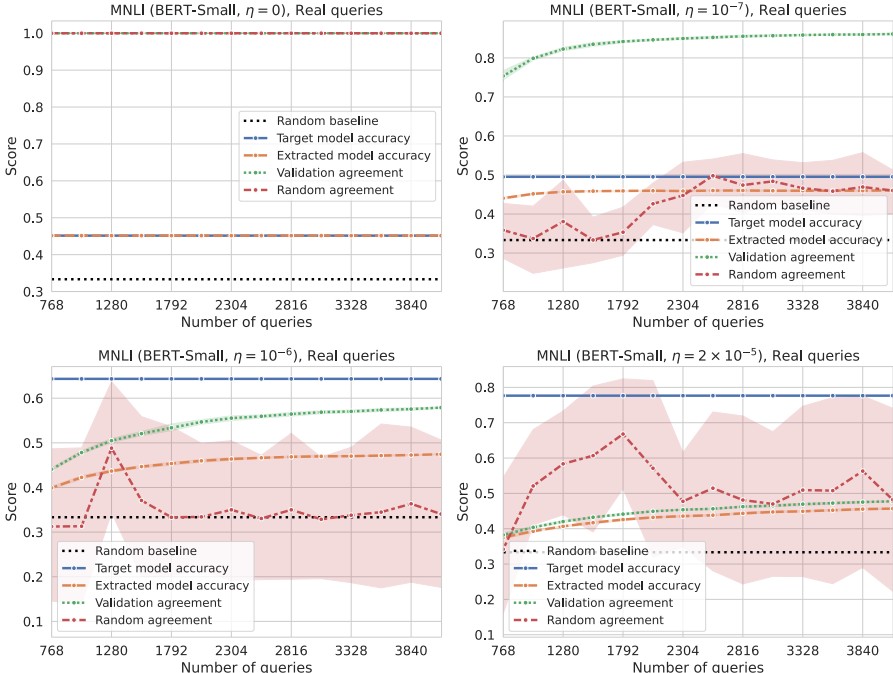

Figure 11: Extraction with in-distribution queries

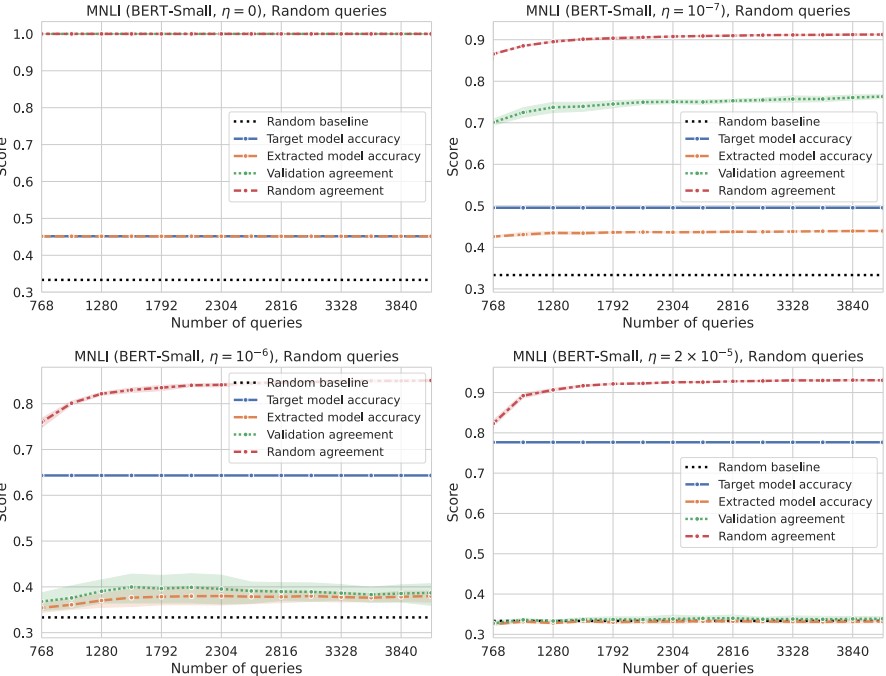

Figure 12: Extraction with random queries

