# OpenReview forum: "Grey-box Extraction of Natural Language Models"
_ICLR.cc/2021/Conference — Reject_

### Official Review · AnonReviewer2 · 2020-10-16
**Interesting research questions and thorough analysis, but attacks are weak in practical settings**

**Rating:** 4
**Confidence:** 5

**Review:**

Summary: This paper is an interesting study of algebraic model extraction attacks on modern NLP models based on BERT. Model extraction is the setting where a malicious attacker tries to reconstruct a copy of a black-box inference API without access to the original training data. Prior work [1] showed these attacks are possible on BERT models using a distillation-like learning method, using gibberish sequences of words as queries to the API. However, these attacks needed large number of queries for success. This work adopts a different strategy --- equation solving the parameters of the neural network using least square linear algebra methods. This not only allows extraction with lesser queries, but also ensures greater similarity between the API and extracted model ("high fidelity", [2]). The attacks in this paper work perfectly in settings where BERT is frozen and a single classification layer is fine-tuned. However, the attacks are not as effective in the more practical setting where BERT is fine-tuned, and the authors perform a thorough analysis varying critical hyperparameters.

-----------------------------------

Strengths of the Paper:

1. This is a new attack setup (especially in the BERT fine-tuning setup), algebraic attacks have only been attempted on very shallow neural networks with ReLU activations. Algebraic attacks have several advantages for the attacker like high fidelity and small query budgets. In the frozen BERT, single layer setting this works perfectly with a very small query budget (however, see my Weakness #1).

2. The paper is well written and easy to understand, and authors do a very good analysis of their attacks varying important hyperparameters like learning rate, number of queries, type of queries.

-----------------------------------

Weaknesses of the Paper:

1. I don't think the setting where the attack works perfectly (frozen BERT with a single classification layer) is practical. Theoretically it's fairly obvious this should work, and I think the main contribution here is a empirical confirmation that it works with real data. There are a number of reasons why this is not practical --- (1) there are actually 2 layers between the sequence_output and the final logits, with a tanh activation (see https://github.com/google-research/bert/blob/master/modeling.py#L219-L232), or look for `BertPooler` in the HuggingFace code. These two layers are needed to separate the MLM representation from the logits. Even in the frozen setting, I anticipate fine-tuning atleast these two classification layers; (2) target accuracies are quite poor without fine-tuning. 75% on SST2 (Target Acc from Table 1) is quite poor, even a 1-layer CNN does much better and gets 83-88% accuracy (https://arxiv.org/abs/1408.5882). Similarly, the Target Acc. for MNLI is close to 33%. Without fine-tuning and just a single classifier layer, I don't expect people to use BERT; (3) Finally access to probability distributions / logits might be a strong assumption in structured prediction NLP tasks like question answering or NER.

2. In the more practical setting of finetuning the model, the attacks are not effective. While I like the overall idea of leveraging the BERT pretrained checkpoint to do algebraic attacks, the authors' results show that this by itself is not sufficient to make an effective attack. The authors statement "For the fine-tuned models, agreement drops to 10% below the learning-based state-of-the-art" is not entirely correct. It is only true on the simpler SST-2 task, where even 1-layer CNNs perform exceedingly well. In the harder MNLI task, agreement is far lower than state-of-the-art [1], with a gap of 44% vs 82.2%. Performance of the extracted models on MNLI are quite low, about 40-45% in Figure 1 which is quite close to random guessing (BERT-base gets 84-85% accuracy).

-----------------------------------

Overall Recommendation:

The authors did a good job with presentation and studied an interesting algebraic attack. However, the attack only works in an impractical setting of a frozen BERT, and is ineffective in the more practical setting of finetuning BERT. Intuitively, it's fairly obvious this attack should work in the frozen BERT, single layer setting. This result by itself is not sufficient for acceptance to ICLR. While I'm leaning reject, I encourage the authors to explore the BERT fine-tuning setting more. For instance, can a hybrid attack be constructed which uses the best of both worlds? Since queries do not seem to cost much [1], can the attacks be stronger in this hybrid setting with more liberal query budgets?

-----------------------------------

Minor Issues:

In Proposition 1, uniformly sampling from a n-d cube is not entirely correct. BERT has a fixed discrete input space, since you only feed text as input to BERT. You are going to have a maximum of V^L unique points in the support of the [CLS] vector space (where V is the vocab and L is the maximum sequence length). Since V ~ 30k and L = 512, I guess it's not a problem practically.

392.702 ---> 392,702

-----------------------------------

References:

[1] - https://arxiv.org/abs/1910.12366
[2] - https://arxiv.org/abs/1909.01838

---

> ### Author Response · Authors · 2020-11-23
> **Response to AnonReviewer2**
>
> ## Hybrid attacks
>
> We hinted in the Discussion section in the submission at ways of combining both classes of attacks. Motivated by the Reviewer's questions we investigated this further and can report some positive results:
>
> We ran a hybrid attack that first uses the (black-box) distillation approach from [1] to get a better approximation of the target model. We use the same hyperparameters as [1]: train for 3 epochs, batch size 32, learning rate 3e-5, Adam optimizer with decoupled weight decay (AdamW in PyTorch) and cross-entropy loss on soft labels. We use the resulting model instead of the pretrained base model in our algebraic attack to extract the classification layer, reusing the same queries as for distillation.
>
> When using a dataset ~10% the size of the training dataset of the target model, and targeting a model fine-tuned with a high learning rate (2e-5), the learning-based attack already performs well, but running our algebraic attack on top of it consistently results in a gain of around 1% in both accuracy and agreement. Given that no extra queries are required and the cost of the algebraic attack is negligible, this is a surprising gain.
>
> For example, for BERT-Base fine-tuned on MNLI with learning rate 2e-5, using 32,768 queries distillation results in 79.03% accuracy and 86.98% agreement. The algebraic attack improves these results to 80.20% accuracy and 88.34% agreement. We included more results in Appendix A in the rebuttal revision. We will perform a more thorough evaluation in a later revision.
>
> We think that algebraic and learning-based attacks can be complementary. Our experiments show that algebraic attacks outperform learning-based attacks when the target model's parameters are trained with a low learning rate, or when only a few queries are available. Our experiments also explore the limits of algebraic attacks and show that in some other scenarios learning-based attacks clearly perform better.
>
>
> ## Frozen encoder layers
>
> We concur with the reviewer that it is in general undesirable to freeze all layers of a pretrained model during fine-tuning. This setting is particularly unfavourable for modestly sized models like BERT-Small and BERT-Base in the tasks that we consider, where simpler models do better. The goal of our experiments in this setting is indeed to confirm empirically that the theory behind the attack holds up when using real world models and data.
>
> We note, however, that freezing layers of a pretrained model or fine-tuning them with a learning rate much lower than the one used for task-specific layers is not that uncommon in practice. For instance, the documentation of the HuggingFace Transformers library describes how to freeze all layers of a pretrained BERT model during fine-tuning (https://huggingface.co/transformers/training.html#freezing-the-encoder). The sample code provided freezes the BERT encoder layers **and** the `BertPooler` layer. Searching in GitHub and StackOverflow shows that this is a recurrent request from practitioners (e.g., https://github.com/huggingface/transformers/issues/400, https://github.com/google-research/bert/issues/637). As another example, the Keras developer guide on transfer learning and fine-tuning recommends initially freezing all layers of a pretrained model (Keras provides facilities for doing this) and only later optionally fine-tuning them using a very low learning rate (https://keras.io/guides/transfer_learning/).
>
> The folklore that earlier layers of a model extract universal features while later layers perform task-specific modelling, together with computational resource considerations, also make it attractive to freeze earlier layers. This might be because training all layers is infeasible, or because one wants to share parameters between several downstream tasks. Although Adapters achieve a higher degree of parameter sharing, freezing early layers and only fine-tuning top layers is a simpler solution to share parameters between downstream tasks (https://arxiv.org/abs/1902.00751). Another recent evaluation shows that Transformer models do comparably well when only some final layers are fine-tuned (https://arxiv.org/abs/1911.03090).
>
> We observed empirically that either freezing early layers of a model or training them with a lower learning rate improves performance of algebraic attacks.
>
> ## Effectiveness
>
> The reviewer is right that the 10% drop compared to learning-based attacks only holds for the SST-2 task. We corrected our claim in the paper.
>
> Our early results on hybrid attacks suggest they are effective in the practical setting of fine-tuning BERT with typical hyperparameters.
>
> ## Minor issues
>
> Good observation. We added a footnote in Proposition 2 to explain that although a model with vocabulary $V$ and maximum sequence length $L$ can only produce $|V|^L$ different embeddings, in practice this is no more problematic than using finite-precision arithmetic (where orders of magnitude fewer numbers are representable).

---

> > ### Comment · AnonReviewer2 · 2020-11-25
> > **Thanks for your replies!**
> >
> > Thank you so much for the detailed rebuttal, I really appreciate it. Here are my thoughts,
> >
> > > hybrid attacks
> >
> > This looks like a promising direction! But how did you get 79% accuracy with just 10% data on BERT-base? Based on Table 3 in [1], this is only possible if you use real NLI data (the numbers in Table 3 of [1] use BERT-large and get about 81%, but are much lower for WIKI and RANDOM at 0.1x). What was the 10% data used? How does it work with the RANDOM / WIKI strategies of [1]?
> >
> > I completely agree that algebraic attacks are promising with a small query budget, and many experiments in [1] worked well only with liberal query budgets (which may not be practical for an attacker). I encourage the authors to continue working in this direction.
> >
> > > frozen encoder layers
> >
> > While I agree freezing BERT may not be a bad idea for computational reasons, my concern is having just a one-layer network on top of it with a softmax nonlinearity. The success of word embeddings and ELMo (pre-BERT) all came by much deeper networks on top of frozen pretrained representations. As your paper showed, just training one layer is not a good idea to optimize performance (might as well use a non-BERT model like the CNN from [2]).
> >
> > **Overall**: I really appreciate the rebuttal and the efforts put into conducting the hybrid attacks. I think they are a promising direction. However, I think more work is needed before this paper is ready for publication. The current set of experiments will not be sufficient for me to increase my score to acceptance.
> >
> > [1] - https://arxiv.org/pdf/1910.12366.pdf
> > [2] - https://arxiv.org/pdf/1408.5882.pdf

---

> > > ### Author Response · Authors · 2020-11-25
> > > **Thanks for following up!**
> > >
> > > ## Hybrid attacks
> > >
> > > > What was the 10% data used?
> > >
> > > We used real NLI data. We combined the `test_matched`, `test_mismatched`, `validation_mismatched` datasets of MNLI and enough points from the `train` dataset (3,293, precisely). We evaluate accuracy and agreement on the `test_matched` dataset, which is disjoint from the ones we used for extraction.
> > >
> > > > How does it work with the RANDOM / WIKI strategies of [1]?
> > >
> > > For an easier comparison to [1], we re-run this experiment using extraction datasets generated with the RANDOM and WIKI strategies (using code from https://github.com/google-research/language/tree/master/language/bert_extraction/).
> > >
> > > For example, for BERT-Small fine-tuned on MNLI with learning rate 2e-5, using 32,768 queries generated with the WIKI strategy, distillation achieves 61.63% accuracy and 66.87% validation agreement. Running our algebraic attack on top of it achieves 61.57% accuracy and 67.11% validation agreement.
> > >
> > > ## Frozen encoder layers
> > >
> > > Following the method described in [3], we also ran preliminary experiments in which only some of the layers of the BERT encoder are frozen and we repeated our algebraic extraction of the classification layer. Specifically, for BERT-Base (12 layers) we froze the embedding layer and the first 9 layers (as this was shown in [3] to give good performance) and trained the remaining layers and the classifier with learning rate 2e-5.
> > > For SST-2, this resulted in 89.45% target model accuracy. Algebraic extraction using 1,536 queries achieves 81.77% accuracy, 85.43% agreement on the validation dataset, and 88.87% agreement on random queries. Compared with Figure 2 in our paper, a similar target model accuracy is achieved with a learning rate of 1e-6, but our extracted model accuracy improves by 1%.
> > > For MNLI, this resulted in 79.71% target accuracy, 41.10% extracted model accuracy, 42.54% agreement on the validation dataset and 54.48% agreement on random queries. This corresponds to a point between 1e-6 and 1e-5 learning rate in our Figure 2.
> > > We will perform further experiments freezing different numbers of base layers to explore this further.
> > >
> > > > I encourage the authors to continue working in this direction.
> > >
> > > For the next revision, we will run  the following experiments and integrate the results into the paper body:
> > > - We will run experiments with GPT-2 in addition to BERT, and multi-label classification tasks beyond MNLI: e.g., Yelp reviews star rating (5 classes), 20Newsgroups (20 classes).
> > > - We already ran preliminary experiments with selective freezing of layers following [3]. We will run more thorough experiments freezing layers gradually and fine-tuning the rest using different learning rates.
> > > - We will run thorough experiments using hybrid attacks for different base models and tasks, varying the learning rate, number of frozen layers, number of queries, and type of queries (i.e., NLI, Random, Wiki).
> > >
> > > > More work is needed before this paper is ready for publication.
> > >
> > > We welcome suggestions for directions to explore beyond those we describe above.
> > >
> > > [3] https://arxiv.org/abs/1911.03090

---

### Official Review · AnonReviewer1 · 2020-10-27
**The paper considers a parameter estimation for the logistic regression and - no surprise - succeeds in it**

**Rating:** 3
**Confidence:** 4

**Review:**

##########################################################################
Summary:

The paper considers the reconstruction of the last layer for NLP data processing models. This problem is equivalent to the parameter estimation for logistic regression in the first of the paper and quite close to it in the second part when we purposely change the encoder via transfer learning.

No surprise, that the reconstruction in this setting works well. This is what we already know from linear algebra and Gauss-Markov [1], Bernstein-von-Mises like theorems in statistics [2, chapter 10].
More interesting is the part about what is happening, when we deal with reconstruction under a transfer learning setting. In this case, we observe a predictable degradation of the quality of the models, but nothing more specific

##########################################################################
Reasons for score:

I vote for rejection, as this paper doesn't contribute to our understanding of what is happening in real-world NLP models with many layers, rather focusing on the last layer fine-tuning.

##########################################################################
More detailed review:

################
Theoretical results

All proposition in the paper are obvious and also equivalent to the recovery procedure for the coefficients of a multiclass logistic regression:
1. Proposition 1 is obvious
2. Proposition 2 is obvious
3. Proposition 3 is obvious
4. Proposition 3 is obvious

The general statement that concludes this section and leads to further experiments should be compared to theoretical results for softmax (or multinominal) regression, see e.g. [3] for some details on the quality of the estimates in this setting. Also, see similar results for logistic regression in [4]. Both these papers present result on the quality of parameters' estimates in a more advanced subsampling setting, and even in this case, they provide the speed of converges for the error of parameter estimates.

So for the benefit of the quality of the paper, I suggest dropping all theoretical results as they are not new.

################
Practical results

1. Due to the reasons similar to that mentioned above the experiments for $\eta = 0$ can be dropped to avoid confusion from the reader
2. For the setting with the fine-tuning of the models, we can see from experiments that after learning emerges a disagreement between the parameters estimates via the proposed procedure and the initial values of parameters. In particular, how can we measure the distance between two models even if they are one-layer logistic regression models, and can we do something if there is one layer in a setting closer to the white box problem.

[1] Henderson, C. R. (1975). Best linear unbiased estimation and prediction under a selection model. Biometrics, 423-447.
[2] Van der Vaart, A. W. (2000). Asymptotic statistics (Vol. 3). Cambridge university press.
[3] Yao, Y., & Wang, H. (2019). Optimal subsampling for softmax regression. Statistical Papers, 60(2), 235-249.
[4] Wang H, Zhu R, Ma P (2018b) Optimal subsampling for large sample logistic regression. J Am Stat Assoc 113(522):829–844

---

> ### Author Response · Authors · 2020-11-23
> **Response to AnonReviewer1**
>
> ## Theoretical results
>
> The problem of extracting models where only task-specific layers are fine-tuned (scenario A) is closely related, but not equivalent, to parameter estimation for regression (scenario B). The key difference is that for (A) the goal is to recover a fixed but unknown set of parameters (i.e. a ground truth) with a minimal amount of data, whereas for (B) one wants to find the best parameters to fit all the available data. In particular, subsampling is not a direct fit for (A) because
> -	we cannot directly choose inputs to the classifier, only to the encoder whose outputs are the inputs of the classifier.
> -	any linearly independent set of embeddings will allow to achieve (A).
> We added a discussion on  the relationship between (A) and (B) to the related work section.
>
> Section 2 is simply intended to explain the methodology of our attack. We use propositions to structure the presentation, but we do not claim that any of the propositions in isolation are novel (or indeed non-obvious to expert practitioners). For instance, we explicitly mention that Proposition 1 is standard. Our contribution is to point out that the combination of these propositions entails a scenario where an algebraic attack is likely to succeed in practice. This attack scenario is novel and has not yet been explored. In Section 3, we empirically show that the attack works not only in the *clean-room* case when $\eta=0$, as one can reasonably hypothesize, but also beyond, which is more surprising.
>
> ## Practical results
>
> 1.	We include the $\eta=0$ case as a *clean-room* illustration of the theory described in Section 2. We also believe it has practical relevance. See our response to Reviewer 2 for a more detailed description of cases where freezing some layers of a pretrained model is beneficial.
>
> 2.	The question of how to measure the distance between two models is very interesting (even in the white-box case for one-layer logistic regression models). We compared our parameter estimates for the extracted layer against the target parameters using L∞ distance. This simplistic metric is a good predictor of how an attack performs when varying hyperparameters for a fixed target model, but it is not meaningful when comparing results across different target models where the magnitudes of parameters can vary widely (as shown in Table 2, even with relatively large L∞ distances, an attack can still do well). Agreement (on in-distribution and out-of-distribution inputs) is a much more meaningful metric that directly aligns with the *high-fidelity* goal of algebraic extraction. It would be interesting to explore this question further. For instance, measuring cross-entropy of soft labels rather than agreement on hard labels could be informative.

---

### Official Review · AnonReviewer3 · 2020-10-29
**Practical method for extracting semi-private language models with some demonstrated success**

**Rating:** 7
**Confidence:** 4

**Review:**

The paper proposes an algebraic attack for extracting the parameters
of a semi-private language model that consists of a pre-trained
encoder and a privately trained classification layer.

The method is to first sample from the input space, compute their
embeddings using the known encoder, and then use the embeddings and
the queried classifier softmax output to solve for the classifier weights.
It overcomes the obstacles encountered by former such attempts due to
the requirements of known embeddings and raw logits.

The paper provides support for the method in arguing that a random
basis (like an arbitrary set of embeddings obtained from encoding a
set of arbitrary, distinct input) is sufficient to serve as a basis
that spans the classifier layer's input space, and that using the
softmax output instead of raw logits can lead to equivalent solutions
up to a translation invariance.

Experiments on two public datasets and two versions of the BERT model
show the effectiveness of the method, and demonstrate that
the number of queries needed is relatively small,
the probes can be drawn from the distribution of legitimate input,
and that fine tuning the encoder makes the attack less effective as the true
embeddings deviate from those computed from the publicly known
encoder.

The paper is well written and the method is sound and practical.
Suggestions on defenses against such attacks are of good reference
value.

One question is whether the proposed approach could be put to some
positive use, such as learning about a model's potential weakness in
the input space?

---

> ### Author Response · Authors · 2020-11-23
> **Response to AnonReviewer3**
>
> ## Positive use
>
> One of the goals of our experiments was to understand how algebraic attacks perform using different types of queries, i.e., in-distribution versus random. We observe that in-distribution queries are more effective and hence it is harder to detect and defend against the attack by observing only the query patterns in the input space. As a next step towards learning a model's potential weakness, we plan to:
>
> 1.	Analyse the embedding space (instead of the input space) for random and in-distribution queries as suggested by Reviewer 4.
> 2.	Explore if we can identify inputs whose embeddings are less affected by fine-tuning and can be used for more effective extraction.
>
> There is an intriguing connection to solutions to *catastrophic forgetting* (aka catastrophic inference), where the goal is to preserve knowledge about previously learned tasks. A pre-trained encoder model that is more robust to catastrophic forgetting will preserve the embeddings of out-of-distribution queries better when fine-tuned, increasing its susceptibility to algebraic extraction attacks.

---

### Official Review · AnonReviewer4 · 2020-10-31
**Important theoretical + empirical results for model extraction attacks, which is helpful and insightful for general NLP interpretability/probing work as well.**

**Rating:** 5
**Confidence:** 4

**Review:**

Summary:

This paper proposes a range of algebraic model extraction attacks (different from the prevalent learning-based approaches) for transformer models trained for NLP tasks in a grey-box setting i.e., an existing, public, usually pretrained encoder, with a private classification layer. Through attacks on different sizes of models and a range of downstream tasks, they observe that only a portion of the embedding space forms a basis of the tuned classification layer’s input space, and using a grey-box method, this can be algebraically computed. The pretraining-finetuning experiments on different tasks also show the smallest number of dimensions needed for high-fidelity extraction, and also that the model extraction attacks effectiveness decreases with fine-tuning the larger models base layers---which is an insight that is very useful for a lot of interpretability/probing work.


Reason for score:

I think this paper is very well-formulated---both theoretically and empirically with promising results that will be useful not just for grey-box adversarial attacks, but also for works interesting in the effects of pretraining-finetuning (which at this point encompasses nearly all NLP tasks). The empirical results look promising---however I would like to see this demonstrated on more than just 2 datasets (and maybe even a GPT-like model, instead of just BERT) to see if (1) the results hold empirically and (2) if there any insights to be gleaned about adversarial attacks from different task structures and model types.


Positive points + questions:

1. The transformation of the raw logits for recovering information is really interesting. In the experiments for the random set of n embeddings chosen to form a basis of the last layer’s input space---are there any insights on what those embeddings amount to semantically; and also what a ground truth selection of embeddings (e.g., that an oracle adversary would compute) should be? It would be helpful to have a discussion and examples of those.

2. Is there a difference in extraction results when using in-distribution queries vs. random? Most of the results say “extraction is possible with both” which is good to see, but a more finer-grained analysis/explanation of benefits/pitfalls of each would really help clarity.

3. It’s nice that both a single-sentence and pairwise-sentence (SST-2 vs. MNLI) task are used to evaluate effects for the fine-tuning experiments in big transformer models.

4. The results look very promising and these insights are extremely helpful even for general probing/interpretability works (especially the learning rate finetuning effects) and also hold up to existing BERT-finetuned results.

5. Unlike previous work, this algebraic model extraction words even with non-linear activation layers---and this is helpful given the current standard of fine-tuning large transformer models e.g., with simple MLP/softmax classifiers.

6. Slightly different from previous work, not only can this work when attacks require embeddings to be chosen, but also when selecting (e.g., random/or from a distribution) needs to be done as well.


Negative points + questions:

1. For the fine-tuning/learning rate experiments it would be good to evaluate this on more than just 2 tasks (e.g., maybe a range of different tasks in GLUE) not only to see if the trend still holds, but also to see if task “type” or characteristics of the task/fine-tuning affect the extraction fidelity.

2. The *extracted model accuracy of BERT-base with MNLI seems to be quite static (almost no effect on increasing or decreasing learning rate)---and it would be really helpful to see how statistically significant those results are and what they look like over different seeds.

3. Is there a comparison between the algebraic approach and a learning-based approach for the same tasks? (I think the paper is novel and useful enough in itself, but it would be helpful to see a side-by-side comparison).

4. Is there a comparison between extracting only a single layer or going beyond to having multiple layers of target/finetuned classifiers? Is this approach feasible and similarly beneficial as a grey-box attack in that scenario? It would be really helpful to have a discussion on what that would require for future work.


Additional minor comments:

This is really well written and placed in literature, no minor nitpicks re: writing!

---

> ### Author Response · Authors · 2020-11-23
> **Response to AnonReviewer4**
>
> ## Positive points + questions
>
> > Is there a difference in extraction results when using in-distribution queries vs. random?
>
> We added full experimental results for extraction with in-distribution and random queries in Appendices B and C. Models extracted using in-distribution queries show better agreement with the target model on in-distribution inputs than on random inputs, with the gap closing as the fine-tuning learning rate increases and agreement on both types of inputs decreases. Models extracted using random queries show better agreement with the target model on random inputs than on in-distribution inputs but the gap does not close as the fine-tuning learning rate increases, with agreement on random inputs remaining high. See, e.g., Figures 5 and 6 in Appendix C in the rebuttal revision.
>
> ## Negative points + questions
>
> > For the fine-tuning/learning rate experiments it would be good to evaluate this on more than just 2 tasks (e.g., maybe a range of different tasks in GLUE) not only to see if the trend still holds, but also to see if task “type” or characteristics of the task/fine-tuning affect the extraction fidelity
>
> We plan to add results for other language classification tasks (with a larger number of classes) as well as for GPT-2-Small in a later revision.
>
> > The extracted model accuracy of BERT-base with MNLI seems to be quite static (almost no effect on increasing or decreasing learning rate)---and it would be really helpful to see how statistically significant those results are and what they look like over different seeds.
>
> We performed a more thorough evaluation (including additional experiments) to answer this question. For each combination of hyperparameters (#queries, learning rate, random vs in-distribution queries, model type), we repeated the attack with 5 different random seeds. The variability is visible in the bands around the curves depicting average model accuracy and agreement in the full results in Appendix B and C and in Figures 1 and 2 in the body of the rebuttal revision.
> The accuracy of models extracted from BERT-Base fine-tuned on MNLI indeed does not vary much with the learning rate, contrary to agreement on random and in-distribution queries. This reflects the *high-fidelity* goal of algebraic extraction attacks, which prioritize agreement over accuracy.
>
> > Is there a comparison between the algebraic approach and a learning-based approach for the same tasks? (I think the paper is novel and useful enough in itself, but it would be helpful to see a side-by-side comparison).
>
> We ran additional experiments comparing the algebraic and learning-based approaches, as well as a hybrid approach that combines both. We report the results in Table 3 of Appendix A.
>
> > Is there a comparison between extracting only a single layer or going beyond to having multiple layers of target/finetuned classifiers? Is this approach feasible and similarly beneficial as a grey-box attack in that scenario? It would be really helpful to have a discussion on what that would require for future work.
>
> Our grey-box attack could be extended to extract multiple task-specific layers by generalizing the attack from https://arxiv.org/abs/2003.04884. This is in principle possible when the added layers form a piecewise-linear network (e.g., a MLP with ReLU activations). The generalization seems feasible when the encoder is frozen and the inputs to the piecewise-linear network are known. It would be very challenging to further generalize the method when the encoder layers are fine-tuned. Specifically, the method relies on testing hypotheses by observing the failure or success of extracting weights. This would be unreliable when the inputs to the piecewise-linear component are not known with certainty.

---

### Author Response · Authors · 2020-11-23
**Summary**

We sincerely thank all the reviewers for their thoughtful feedback!

We summarize the changes we have made in the updated PDF after the rebuttal and briefly describe the additional experiments here. We address technical questions in comments to individual reviews.

-	We performed 5 runs of our experiments for random and in-distribution queries using different random seeds, for all hyperparameter combinations. The full experimental results (including variability in runs) are given in Appendix B for varying learning rate and Appendix C for varying number of queries.

-	We ran additional experiments that compare algebraic and learning-based extraction attacks, as well as a hybrid approach combining both attacks (following reviewers’ suggestions). We report the results in our comments to individual reviews and present them in Table 3 in Appendix A. We will include a more thorough evaluation in a later revision.

---

### Decision · Program_Chairs · 2021-01-07
**Final Decision**

**Decision:**

Reject

**Comment:**

After discussion with the reviewers, it seems that a. without fine-tuning the result is close to being trivial (as noted also by two reviewers) b. with fine-tuning results are lower c. The setup of just a linear classification layer is less common (but exists) d. The cases where extraction succeeds the performance is low such that BERT would not even be used.

In response, the authors offer many interesting directions: a. Propose a new hybrid approach that combines learning-based and extraction-based methods b. Run experiments to try and support the claim that their setup of one linear layer with frozen layers is practical.

These proposed modifications are interesting and show that there is potential in this paper, but it deviates substantially from the original paper and still, caveats remain, so my recommendation is to re-submit after further pursuing the new directions proposed in the response.